Identification of Cassiopea sp. in Lake Macquarie, Australia and revision of the taxonomic status of Cassiopea maremetens Gershwin, Zeidler & Davie, 2010 (Cnidaria: Scyphozoa: Cassiopeidae)

http://orcid.org/0000-0002-5205-7629 Rowe Claire E. 1 Claire.Rowe@australian.museum
Ahyong Shane T. 1 2
http://orcid.org/0000-0001-9472-8710 Figueira Will F. 3
Burghardt Ingo 1 4
http://orcid.org/0000-0002-1754-6750 Keable Stephen J. 1
1 Australian Museum Research Institute, Australian Museum , Darlinghurst, New South Wales , Australia
2 School of Biological, Earth & Environmental Sciences, University of New South Wales , Kensington, New South Wales , Australia
3 School of Life and Environmental Sciences, University of Sydney , Sydney, New South Wales , Australia
4 Aquatic Ecology, Sydney Water , West Ryde, New South Wales , Australia
Parra Olea Gabriela
Electronic publication date: 2025 Jul 18
Publication date: 2025
Volume: 13
Electronic Location ID: e19669
Received 2025 Jan 24; Accepted 2025 Jun 6
Copyright: © 2025 Rowe et al.
Copyright year: 2025
Copyright holder: Rowe et al.
License: This is an open access article distributed under the terms of the Creative Commons Attribution License, which permits unrestricted use, distribution, reproduction and adaptation in any medium and for any purpose provided that it is properly attributed. For attribution, the original author(s), title, publication source (PeerJ) and either DOI or URL of the article must be cited.
License URL: https://creativecommons.org/licenses/by/4.0/

Keywords: Invasive, Jellyfish, Taxonomy

Funding: Australian Museum Research Institute Postgraduate Award This work was funded by the Australian Museum Research Institute Postgraduate Award. There was no additional external funding received for this study. The funders had no role in study design, data collection and analysis, decision to publish, or preparation of the manuscript.

==============================
Scyphozoans of the genus Cassiopea are notable for their unusual benthic habit of lying upside-down with their exumbrella resting on the substrate and oral arms facing upwards resulting in their common name “upside-down jellyfish”. Cassiopea includes species that have been historically confused because of taxonomic ambiguity. Additionally, some species are considered to be invasive, which can have significant economic and environmental consequences by impacting fisheries, tourism, and trophic structures. In temperate southeastern Australia, Cassiopea medusae were first reported in temperate Wallis Lake and Lake Illawarra in 2016, and then Lake Macquarie in 2017, though historically these jellyfish have a more northern tropical distribution in Queensland, eastern Australia. Owing to the invasive potential of Cassiopea, correct species identification is crucial for future management. To address this knowledge gap, this study used genetic comparison through the cytochrome c oxidase subunit I (COI) barcoding gene and morphometric analysis, together with revision of type and topotype material of Cassiopea maremetens Gershwin, Zeidler & Davie, 2010, an incompletely known nominal species from Queensland, to investigate the identity of Cassiopea occurring in Lake Macquarie. The morphometric analysis was also used to identify key features that distinguish the Lake Macquarie species from a second species, designated Cassiopea sp.3, that is also expanding its range southwards in eastern Australia, and which may be sympatric in some areas. The results of this study show the species occurring in Lake Macquarie is Cassiopea xamachana Bigelow, 1892, originally described from Jamaica and subsequently widely reported from the Western Atlantic and the Indo-West Pacific. Additionally, we demonstrate that Cassiopea maremetens, is a junior synonym of C. xamachana. Morphological characters that can be most readily used to distinguish mature specimens of C. xamachana from C. sp.3, which has an overlapping distribution on the Australian east coast, are: (1) the number of large appendages on the oral disc, which is much higher in Cassiopea sp.3 (at least 1 but up to 14) vs. a maximum of two in C. xamachana; (2) the oral arm branching pattern, which is usually alternating for C. xamachana, but a combination of alternating, bifurcating and pinnate for Cassiopea sp.3; (3) the length of the large appendage on the oral arm, which is proportionally longer relative to the bell diameter in C. xamachana.

Introduction

Upside-down jellyfish (Cassiopea spp.) are unusual scyphozoans because they spend the majority of the medusa phase of their lifecycle resting on the benthos, with the bell facing downwards, and the oral arms directed upward into the water column (Ohdera et al., 2018). Cassiopea occurs world-wide in tropical to sub-tropical regions, in shallow and protected habitats, such as coral reefs, mangrove forests, seagrass beds, or even human-made environments (Ohdera et al., 2018; Rowe et al., 2022b; Stoner et al., 2014; Thé et al., 2020a). Some species of Cassiopea, however, are considered to have been artificially translocated, and in some cases becoming invasive (Bolton & Graham, 2006; Holland et al., 2004; Keable & Ahyong, 2016; Rowe et al., 2022b). This is especially the case for Cassiopea andromeda (native to the Red Sea), which has been recorded from shrimp farms in Brazil (Thé et al., 2020b), and the eastern Mediterranean Sea as a Lessepsian migrant where it is common on sandy bottom coves on the Aegean coast of Turkey (Gülşahin & Tarkan, 2012), and other harbours in the Mediterranean (Maggio et al., 2019). Range expansions can occur via localised dispersal, or through direct anthropogenic interventions including shipping, ballast water and release through the aquarium trade (Bolton & Graham, 2006; Graham & Bayha, 2008; Morandini et al., 2017). In some localities, Cassiopea has thrived and become invasive (Thé et al., 2020b). Characteristics considered important in their success as invaders include tolerance of a broad range of environmental parameters, ability to be mixotrophic, high reproductive rate and the ability to reproduce both sexually and asexually (Holland et al., 2004; Keable & Ahyong, 2016; Maggio et al., 2019; Mammone et al., 2023; Morandini et al., 2016; Schiariti et al., 2014; Thé et al., 2023). Additionally, Cassiopea can have reproductive blooms in high densities (Morandini et al., 2017; Stoner et al., 2014). Under increasingly favourable environmental conditions associated with human impacts, including coastal development and rising temperatures with climate change, these blooms may occur more frequently (Brotz & Pauly, 2016; Ohdera et al., 2018; Richardson et al., 2009; Rowe et al., 2022a; Stoner et al., 2014). High densities of Cassiopea can lead to competition with other benthic fauna for light, space and food, and smothering of seagrass and other aquatic vegetation (Stoner et al., 2011).

Cassiopea is the only genus in the family Cassiopeidae. The most recent synopses of the genus recognised 12 valid species (Collins, Jarms & Morandini, 2025; Jarms et al., 2019): 1, C. andromeda (Forskål, 1775); 2, C. culionensis Light, 1914; 3, C. depressa Haeckel, 1880; 4, C. frondosa (Pallas, 1774); 5, C. maremetens Gershwin, Zeidler & Davie, 2010; 6, C. mayeri Gamero-Mora et al., 2022; 7, C. medusa Light, 1914; 8, C. mertensi (Brandt, 1838); 9, C. ndrosia Agassiz & Mayer, 1899; 10, C. ornata Haeckel, 1880; 11, C. vanderhorsti Stiasny, 1924; and 12, C. xamachana Bigelow, 1892. However, species of the genus can be extremely difficult to distinguish based on morphological characters alone because of their conservative morphology, with few known reliable morphological taxonomic characters (Gamero-Mora et al., 2022; Holland et al., 2004). Such a situation is commonplace in scyphozoan systematics, which has traditionally based descriptions on morphological characters that have to be interpreted from soft structures whose form is often susceptible to the vagaries and differences in preservation leading to possibly premature assertions of high morphological plasticity (Arai et al., 2017; Gohar & Eisawy, 1960; Holland et al., 2004). This is especially the case for Cassiopea, where analysis based solely on morphological characters is challenging because morphological differences between species are usually subtle and a number of features vary between stages of growth and possibly habitat (Hopf & Kingsford, 2013; Maggio et al., 2019). As a result, it has been suggested that an integrative approach, using both morphological and genetic data will provide more reliable taxonomic separation (Arai et al., 2017; Dawson, 2003; Holland et al., 2004; Maggio et al., 2019).

Molecular study has often revealed unrecognised species in many marine taxa, suggesting that marine biodiversity is higher than previously thought, and speciation is more frequent than originally recognised (Dawson & Jacobs, 2001; Knowlton, 2000). For example, the scyphozoan, Aurelia aurita (Linnaeus, 1758), long thought to be a single widespread species based on morphology has been revealed, based on molecular data, to be a complex of at least 28 species (Dawson & Jacobs, 2001; Lawley et al., 2021). One of the first molecular studies of Cassiopea focused on Hawaiian fauna using a single gene (mitochondrial cytochrome c oxidase I, Holland et al., 2004), and tentatively concluded that there were six species including: (1) C. frondosa, which originates from the western Atlantic and at that time considered by Holland et al. (2004) to be the only morphologically well characterised species; (2) C. andromeda, initially reported from the Red Sea and subsequently the Western Atlantic; (3) C. ornata, reported from Indonesia, Palau and Fiji; and (4–6) species that were not associated with named species based on their morphological characters. Subsequently, genetic analysis has been used as a tool for identifying species of Cassiopea, such as C. andromeda in shrimp farms in Brazil (Thé et al., 2020b) and from Italy (Maggio et al., 2019), respectively. Additionally, utilising genetic techniques, new species have been identified, including two different lineages from Palau (Cassiopea sp.5 and sp.6, Arai et al., 2017), which were not sequenced by Holland et al. (2004). A recent synoptic phylogenetic study of Cassiopea (Gamero-Mora, 2020) recognised at least 17 species of which only six can be associated with formal published names, the most recently described being C. mayeri and C. culionensis (Gamero-Mora et al., 2022; Gamero-Mora, 2020). The identity of at least six named species is currently ambiguous, indicating significant revision of the genus is required (Gamero-Mora et al., 2022; Gamero-Mora, 2020).

In Australia, Cassiopea has typically been recorded from sub-tropical and tropical latitudes, occurring north of 27°58′S on the east coast and 16°08′S on the west coast, with the exception of an apparently introduced population established near a warm water outlet from a power station at Angas Inlet, Adelaide, South Australia (approximately 34°48′S, 138°32′E, Keable & Ahyong, 2016; Southcott, 1982, Fig. 1). To date, four named species of Cassiopea (C. andromeda, C. maremetens, C. ndrosia, C. ornata) and two undescribed species have been reported from Australia (Gershwin, Zeidler & Davie, 2010; Keable & Ahyong, 2016). Cassiopea maremetens, however, is the only species described from Australia (type locality: Pelican Waters, Queensland, 26°49′42″S, 153°06′48″E); other Australian species were described from elsewhere. To date no genetic data for C. maremetens have been available, but it morphologically closely resembles C. andromeda and C. ndrosia (Gershwin, Zeidler & Davie, 2010; Keable & Ahyong, 2016).

Figure 1 Localities for Cassiopea compared in this study.

Specimens sourced from the field and from museum collections. Map created in R.

Keable & Ahyong (2016) reported a southward range expansion of two species of Cassiopea on the east Australian coast into New South Wales, identifying C. cf. maremetens in Wallis Lake (32°11′45″S, 152°29′56″E) and C. ndrosia in Lake Illawarra (34°31′36″S,150°51′53″E). A further southern population of Cassiopea was discovered in 2017 in Lake Macquarie (33°04′00″S, 151°32′42″E), approximately mid-way between Lake Illawarra and Wallis Lake (Rowe et al., 2022b, Fig. 1). This raises questions about the identity of the species in Lake Macquarie and from where it may have originated. This is especially important because of the invasive potential of Cassiopea and its environmental impacts.

This study aims to determine the identity of the species of Cassiopea occurring in Lake Macquarie using molecular and morphological data, and to re-assess the taxonomic status of C. maremetens in relation to Cassiopea from New South Wales.

Methods

Taxon sampling

Specimens were collected under New South Wales (NSW) Department of Primary Industries permit FP23/41, and the locations and dates are listed in Table 1. The specimens were collected using hand-nets, either from a paddling vessel or by wading into the water. For further details see the Taxonomy section and Tables S1–S3.

Table 1 Location, date, latitude and longitude of where Cassiopea specimens were collected for this study.

Location	Site	Latitude and longitude	Date	
Lake Macquarie, New South Wales	Lake Petite	33°7′3″S, 151°32′6″E	4 June 17	
		9 February 2018	
		9 May 2019	
Kilaben Creek	33°1′43″S, 151°35′1″E	9 May 2019	
Karignan Creek	33°10′32″S, 151°34′0″E	15 May 2018	
		6 May 2019	
Mannering Bay	33°9′28″S, 151°31′41″E	7 May 2019	
Myuna Bay	33°03′59″S, 151°32′43″E	2 December 2019	
Wallis Lake, New South Wales	Pipers Creek	32°11′58″S, 152°30′39″E	18 September 2014	
			28 April 2015	
			8 April 2016	
			10 May 2019	
Port Hacking, New South Wales		34°04′51″S, 151°08′01″E	31 May 2021	
Lake Illawarra, New South Wales		31′14036″S, 150°51′52″E	1 June 2021	
Coombabah Creek, Queensland		27°54′26″S, 153°22′57″E	8 March 2021	
Pelican Waters, Queensland		26°50′01″S, 153°06′44″E	10 March 2021	

Tissue samples from each specimen were dissected from the gonads (extracted from the subgenital pit) and the tip of one of the oral arms; these were fixed and preserved in 95% ethanol. The remainder of the specimen was fixed in 10% formalin-seawater solution to maintain the morphological characteristics. All New South Wales specimens are deposited in the collections of the Australian Museum, Sydney (AM).

Additional genetic and morphological samples

Additional tissue samples were sequenced from specimens made available by the Queensland Museum, Brisbane (QM), South Australian Museum, Adelaide (SAM), and ReefHQ, Townsville (S1). Additionally, available genetic sequences of Cassiopea from Australian and overseas localities were downloaded from GenBank, including two scyphozoan outgroups, Catostylus mosaicus (Quoy & Gaimard, 1824) and Aurelia aurita (Linnaeus, 1758), following Holland et al. (2004) (S1 and S2).

Collections preserved at the AM, QM and SAM were examined for morphological comparison (S3). Due to uncertainty regarding many characters of C. maremetens, this included type material from QM to redescribe the species (see Taxonomy section).

Genetic analysis

DNA extraction, PCR amplification and sequencing

DNA was extracted from gonad or oral arm tissue using an Isolate II Genomic DNA Kit following manufacturer’s instructions. An approximately 500-bp fragment of the mitochondrial gene cytochrome c oxidase subunit I (COI) was sequenced with the primers Lobo Forward (KBTCHACAAAYCAYAARGAYATHGG) and Lobo Reverse (TAAACYTCWGGRTGWCCRAARAAYCA) (Lobo et al., 2013). COI was selected for its utility in distinguishing species in multiple Cassiopea studies (Arai et al., 2017; Holland et al., 2004; Gamero-Mora et al., 2022; Daglio & Dawson, 2017; Morandini et al., 2017; Maggio et al., 2019). Polymerase chain reactions (PCRs) were performed in a total volume of 20 μl with Invitrogen 10x PCR buffer (2 μl), both primers (0.4 μl each), Invitrogen MgCl2 (1.5 μl), dNTPs (1.5 μl), Invitrogen Taq DNA Polymerase (0.1 μl), Milli-Q H2O (13.1 μl) and the sample (1 μl). Reaction conditions for COI were as follows: 94 °C for 3 min; 40 cycles: at 94 °C (30 s), 52 °C (30 s) and 72 °C (60 s) a final extension at 72 °C (5 min). PCR products were assessed using agarose gel electrophoresis stained with GelRed (Biotium), with the inclusion of a 1,000 bp DNA ladder (ThermoFischer). PCR products were sent to Macrogen (Macrogen Inc. Seoul, Republic of Korea) for sequencing.

Molecular data analysis

A total of 38 COI sequences were obtained from various populations of Cassiopea from around Australia for this study (S1). The sequences were viewed and edited using Geneious (V.2020.0.5) and matched against other sequences in the GenBank database using the Basic Local Alignment Search Tool (BLAST) search. All of the mitochondrial COI sequences from populations in Australia obtained in this study and from GenBank were aligned using MEGA10: Molecular Evolutionary Genetics Analysis version 10 (MEGA) (Stecher, Tamura & Kumar, 2020) using the Multiple Sequence Comparison by Log-Expectation (MUSCLE) algorithm (total of 48 sequences, S1). MEGA was then used to compute pairwise distances between sequences using the Kimura 2-parameter model (K2P), followed by the construction of a maximum likelihood (ML) tree using Tamura 3-parameter, which was selected as a result of the lowest Bayesian information criterion (BIC) value when comparing models in MEGA. Bootstrap values were calculated using 1,000 pseudoreplicates implemented in MEGA. Analyses were rooted to Catostylus mosaicus and Aurelia aurita (see Identification of the Lake Macquarie population in Results).

This method was repeated using a sequence from each locality in Australia, compared to populations worldwide, including two sequences originating from Israel and Singapore, respectively, sequenced in this study, and Cassiopea COI sequences available on GenBank (total of 97 sequences, S2, see Comparison of results).

Additionally, a species delimitation analysis was completed using Bayesian implementation (bPTP) of the Poisson tree process model to infer putative species boundaries on a phylogenetic input tree (Zhang et al., 2013). The input tree used in the bPTP analysis was the resulting trees from both the regional and global analysis. The bPTP was run as a rooted tree with 100,000 generations, 10% burn-in and outgroups were removed.

Morphological analysis

Results from the molecular analysis were used to inform specimen sampling for morphometric analysis. A suite of morphological traits were identified and measured following those examined by Morandini et al. (2017), Mayer (1910), Kramp (1961), Hummelinck (1968), Gershwin, Zeidler & Davie (2010), Keable & Ahyong (2016), Jarms et al. (2019). Morphological observations (Fig. 2) were made using a magnification lamp and dissecting microscope. In most cases the exumbrella diameter was measured across the widest point to the nearest millimeter and used to compare size ratios of different features between specimens. However, some specimens have a brittle exumbrella that would not fully open, in which case their condition was noted. Exumbrella diameter, overall height and oral disc height were measured with the oral arms of the specimen lying on top of their exumbrella and with the aboral surface of the exumbrella oriented in the ventral position as when the specimen is naturally at rest on the substrate when encountered in the field (i.e., the typical ‘upside-down’ position). Overall height and oral disc height was measured from the base of the exumbrella and the intersection of oral disc with the bell, to the top of the oral disc. The number of rhopalia and lappets, and the number of lappets per paramere (i.e., between two successive rhopalia, Gamero-Mora et al., 2022) were counted and their shape noted (i.e., round or flat edge, each lappet distinct with grooves between them or connected with a smooth edge, short or long). While the shape and colour of large appendages have been found uninformative in identifying species of Cassiopea in other analyses (Gamero-Mora, 2020), it was observed that their position and size consistently differed between populations occurring in this study on the east coast of Australia. Therefore, the number and shape of large appendages along the oral arms and on the oral disc were recorded. We also measured the length of largest appendage on the oral arm and oral disc from the base to the tip, as well as the width at the midlength. Additionally, the specimen was inverted so that the dorsal surface of the exumbrella was exposed, and the exumbrella was folded back so the oral arms were visible, allowing the length of the oral arms to be recorded from the proximal base at the oral disc to the distal tip of the main trunk. The main oral arm trunk and lateral oral arm branching patterns were also recorded as this is considered to be useful in diagnosing species of Cassiopea (Gamero-Mora, 2020), with three different patterns observed in this study: 1, alternating — the lateral branches switch sides and are separated along the main oral arm trunk; 2, pinnate—the lateral branches are arranged on either side of the main oral arm trunk in pairs opposite each other; 3, bifurcating—the main trunk of the oral arm is divided into two subequally broad distal branches (in this case the distance from the oral disc to the proximal base of the bifurcation, and also whether each branch was equal in length, was recorded).

Figure 2 Schematic diagram of Cassiopea indicating key morphological features examined.

Created with BioRender.com.

Standardisation of body size

To ensure that all count data were not skewed by body size, a regression was completed for each continuous variable against exumbrella diameter to determine if there was a positive relationship. If a significant relationship was detected, the variable was divided by the exumbrella diameter for each individual. If a significant relationship was still detected, the z-score (Curtis et al., 2016) was calculated for each individual using the following formula:

Z=x−meanstandarddeviation

where x is the continuous variable for a specimen and after which, all data was standardised.

Morphometric analysis

To determine if there was any clustering separating putative species based on morphological characters, and if this differed between populations, a non-metric multidimensional scaling (nMDS) was constructed based on a Bray-Curtis similarity matrix using both the continuous data that was standardised for bell diameter, and categorical variables (i.e., oral arm branching pattern; ring canal presence; distribution of large appendages; shape of subgenital pit; and shape of oral arm). This was completed using both the ‘vegan’ package and ‘ggplot2’ in R (v.4.1.1.) (R Development Core Team, 2021). To test for morphological differences a one-way permutation multivariate analysis of variance (PERMANOVA) was completed, with putative species as the fixed factor (3-levels: presumptive C. xamachana, Cassiopea sp.3, Cassiopea sp.2, as listed in Table S3). Additionally, a Ward hierarchical clustering (Cluster) analysis was undertaken to determine morphological similarities between individual samples. Any morphological differences between populations within species for putative C. xamachana and Cassiopea sp.3 were tested using one-way PERMANOVAs with the location of the populations as a fixed factor (3-levels for each species). If there was a significant difference between groups of specimens, a cluster analysis was completed to determine how individuals clustered across populations and in which localities these are morphologically similar.

To establish which features distinguish putative species, a similarity percentage analysis (SIMPER) was applied using the ‘vegan’ package, and the six most influential features were identified.

Results

Genetic analysis

Identification of the Lake Macquarie population

The ML tree (Fig. 3) based on COI of Cassiopea from all sampled localities in Lake Macquarie, Wallis Lake, Pelican Waters, Gold Coast and Moreton Bay formed a strongly supported clade (bootstrap support 100%); all were genetically identical (K2P ± S.E.: 0.00 ± 0.00, Table S4) suggesting they represent a single species, distinct from those of Lake Illawarra (mean between populations K2P ± S.E.: 0.318 ± 0.00), Port Hacking (0.318 ± 0.00), Coombabah Creek (0.324 ± 0.00), and the Northern Territory (0.32 ± 0.00), which were closely related to each other (mean within species K2P ± S.E.: 0.021 ± 0.00). Additionally, the Lake Macquarie specimens differed from those from South Australia (0.078 ± 0.00), Lizard Island (0.294 ± 0.00), and Port Douglas (0.273 ± 0.00). The bTP analysis for species delimitation estimate the presence of six to 36 species with a mean of 21.64. Five species are recognised herein, including Cassiopea from Lake Macquarie, Wallis Lake and Pelican Waters as one species, those from Lake Alexander, Lake Illawarra, Port Hacking and Coombabah Creek as a second species, and each of the populations from Angas Inlet, Port Douglas and Lizard Island, as separate species (see Discussion section and Fig. 1).

Figure 3 Maximum likelihood tree of Australian Cassiopea based on COI gene.

Bootstrap values (1,000 pseudoreplicates) displayed at each node and scale bar indicates length of each branch. Blue indicates sequences from Lake Macquarie specimens. *Material from the type locality of Cassiopea maremetens Gershwin, Zeidler & Davie, 2010.

Comparison of results

The global ML tree (Fig. 4) recovered the Lake Macquarie, Wallis Lake and Pelican Waters specimens in the same clade and genetically invariant, with low divergence from populations in Hawaii, Brazil, Panama and the Florida Keys (mean within species K2P ± S.E.: 0.016 ± 0.00, Table S5), previously identified as Cassiopea xamachana (Holland et al., 2004; Gamero-Mora et al., 2019; Stampar et al., 2020, S2, see Discussion). The low COI divergence suggests that these populations from different localities all represent the same widespread species. Additionally, specimens from Lake Illawarra, Coombabah Creek, Port Hacking and the Northern Territory (herein referred to as Cassiopea sp.3) form a separate clade and are closely related to another species from Japan, Hawaii and Papua New Guinea (Fig. 4, Table S5). The bTP analysis for species delimitation indicates an estimate of 16 to 38 species with a mean of 25.13.

Figure 4 Global phylogeny of Cassiopea based on COI gene.

Bootstrap values (maximum likelihood, 1,000 reiterations) displayed at each node and scale bar indicated branch length. Blue indicates Lake Macquarie specimen sequences. *Material from the type locality of Cassiopea maremetens Gershwin, Zeidler & Davie, 2010.

Morphometric analysis

The nMDS ordination of the specimens of Cassiopea examined is illustrated in Fig. 5. The stress level associated with this two-dimensional plot was 0.176, demonstrating that there were some morphological characters distinguishing the species (Dexter, Rollwagen-Bollens & Bollens, 2018). The nMDS ordination plot formed separated groups based on morphological dissimilarities (Fig. 5). The PERMANOVA confirmed that there are significant morphological differences between species (R2 = 0.395, pseudo-F = 30.089, p < 0.01). Additionally, with the exception of three specimens of Cassiopea sp.3 (AM G.20076 from Port Hacking, G.20060 from Coombabah Creek and AM G.18075 from Lake Illawarra) and two specimens of Cassiopea sp.2 (AM G.17387 and AM G.17370) from Papua New Guinea, the cluster analysis confirmed these results (Fig. S6), and grouped specimens into two morphological groups, with material corresponding to putative Cassiopea sp.3 and Cassiopea sp.2 in the first cluster and presumptive C. xamachana in the second. The PERMANOVA between Australian populations of Cassiopea from Lake Macquarie, Wallis Lake and Pelican waters indicates there are morphological differences between locations (R2 = 0.077, pseudo-F = 2.587, p < 0.05). The follow-up cluster analysis (Fig. S7A) indicated no C. xamachana location clusters driving this significance with the two significant clusters containing specimens from all three locations. However, morphological differences were detected between populations of Cassiopea sp.3 from Lake Illawarra, Coombabah Creek, Port Hacking, and Lake Alexander (R2 = 0.498, pseudo-F = 6.605, p < 0.001). Additionally, the follow up cluster analysis (Fig. S7B) confirmed morphological clusters between locations of Cassiopea sp.2, including cluster 1 containing specimens from the Port Hacking, cluster 2 from Coombabah Creek with two specimens from the Port Hacking, cluster 3 from Northern Territory, with one specimen from Lake Illawarra, and cluster 4 containing specimens from Lake Illawarra.

Figure 5 nMDS plot indicating the clustering between the morphological characters across specimens.

Material examined includes putative Cassiopea sp.3 from Coombabah Creek, Lake Illawarra, Port Hacking and Lake Alexander, Cassiopea sp.2 from PNG and Cassiopea xamachana Bigelow, 1892 from Lake Macquarie, Wallis Lake and Pelican Waters. Stress level of nMDS is 0.176.

SIMPER indicated the three most informative features contributing to separation of putative C. xamachana and Cassiopea sp.3 according to their cumulative contribution are: the number of large appendages on the oral disc; oral arm branching pattern; and length of large appendages on the oral arm (Table 2, Figs. 6–9). Branching pattern is also a distinguishing feature between C. xamachana and Cassiopea sp.2, as well as between Cassiopea sp.2 and Cassiopea sp.3, along with number of ring canals and large appendage distribution (Table 2). The features that were considered less useful when distinguishing between taxa include the shape of the subgenital pit, the number of ring canals, and the shape of the oral arm.

Table 2 SIMPER results showing top three cumulative contributions of morphological characters that distinguish specimens.

Average is the contribution to dissimilarity, Standard deviation of dissimilarity, and Cumulative Sum ordered cumulative contribution.

Comparison of putative species	Morphological feature	Average	Standard deviation	Cumulative sum	Figure	
C. xamachana–C. sp.3	Number of large appendages on oral disc	0.0070329	0.0054158	0.06399	8A	
Oral arm branching pattern	0.0065736	0.0053396	0.12380	9	
Length of large appendages on oral arm	0.0058753	0.0036507	0.17726	8B	
C. xamachana–C. sp.2	Ring canal presence	0.0109182	0.012637	0.08762		
Distribution of large appendages on oral arm	0.0097679	0.006158	0.16601		
Oral arm branching pattern	0.0091168	0.006856	0.23918	9	
C. sp.3–C. sp.2	Ring canal presence	0.011149	0.012803	0.09511		
Distribution of large appendages on oral arm	0.008733	0.005651	0.16960		
Oral arm branching pattern	0.007707	0.005118	0.23534	9	

Figure 6 Number and relative length of large appendages on sampled specimens of Cassiopea.

(A) Number of large appendages on oral disc between. (B) Length (cm) of large appendages on oral arm divided by bell diameter (cm).

Figure 7 Branching pattern of lateral branches along oral arm among sampled specimens of Cassiopea.

Figure 8 Cassiopea xamachana Bigelow, 1892 (holotype of Cassiopea maremetens Gershwin, Zeidler & Davie, 2010), QM-G326486, 17.5 cm diameter).

(A) Oral view of preserved medusa. (B) Oral view of oral disc, red arrow points to large oral appendage located in the centre of the disc. (C) Aboral view of branching pattern on oral arm, red arrow point to large oral appendage. (D) Oral view of the detail of the number and shape of lappets in a paramere. (E) Arrow pointing to rhopalium, oral view.

Figure 9 Cassiopea xamachana Bigelow, 1892, Lake Macquarie, New South Wales, AM-G18732, 19.7 cm diameter, female.

(A) Oral view of preserved medusa. (B) Red arrow points to large oral appendage located in the centre of the oral disc, oral view. (C) Branching pattern on oral arm, red arrow points to large oral appendage, aboral view. (D) Number and shape of marginal lappets. (E) Branching pattern of the oral arm and subgenital pit, oral view.

Discussion

Although the genus Cassiopea is easily recognised, the species have had a confused taxonomic history owing to subtle morphological distinctions and general morphological conservatism in the genus (Holland et al., 2004; Hopf & Kingsford, 2013; López-Figueroa, Stoner & Hallock, 2024; Maggio et al., 2019). This has resulted in a need for molecular data combined with morphology to identify species of Cassiopea in new locations (Holland et al., 2004; Maggio et al., 2019). Additionally, some species of Cassiopea are invasive, with new distribution records around the world, and so correctly identifying the species of Cassiopea is crucial to track and manage their spread (Holland et al., 2004; Keable & Ahyong, 2016; Maggio et al., 2019; Morandini et al., 2017; Thé et al., 2020a).

Sequences of Cassiopea from eastern Australia (Lake Macquarie, Wallis Lake and Pelican Waters) group with, and are indistinguishable at species level from those from Hawaii, Brazil, Palau, Panama, Florida Keys and an additional locality in the northwest Pacific Ocean (Fig. 4). Terminals within this clade are referable to C. xamachana based on genetic comparisons, morphology, and redescription of material that is to be designated as the neotype of the species (Gamero-Mora et al., 2019; Gamero-Mora, 2020; Stampar et al., 2020). However, Cassiopea occurring in Lake Macquarie and Wallis Lake, namely C. xamachana, are also genetically indistinguishable from those from Pelican Waters. This result is particularly significant because Pelican Waters is the type locality of C. maremetens (Gershwin, Zeidler & Davie, 2010). Morphological congruence was also found between specimens from Lake Macquarie and Pelican Waters, including the holotype of C. maremetens. This is the first time that specimens morphologically corresponding to C. maremetens from the type locality of the species have been evaluated based on genetic data.

This provides strong evidence that C. maremetens is conspecific with C. xamachana, with the latter having nomenclatural priority as the older name. This was confirmed by morphological revision of type material of C. maremetens and comparison with additional specimens including topotypes (see Taxonomy and Morphometric Analysis sections). As such, we formally synonymise C. maremetens with C. xamachana, below. Additionally, material from Panama in this clade previously identified by Daglio & Dawson (2017) as C. frondosa appears to be a misidentification and is actually C. xamachana. Conversely, a specimen from the same study identified as C. xamachana and also from Panama was placed here in the C. frondosa clade (Daglio & Dawson, 2017) and is apparently C. frondosa based on the identity assigned in other studies to the members of this clade.

Cassiopea xamachana was first described by Bigelow (1892) from the Caribbean Sea and has subsequently been reported from a wide variety of locations (see Remarks in Taxonomy section). This species has been taxonomically confused with C. andromeda (type locality: Red Sea), both species apparently overlapping in distribution in Florida with a third species, C. frondosa (type locality: Caribbean Sea, Muffett & Miglietta, 2023). Our study supports the results of Gamero-Mora (2020) and Muffett & Miglietta (2023) in recognising C. xamachana as more closely related to C. andromeda (type locality: Red Sea) with a mean pairwise divergence of 7.2%, than to its Caribbean congener, C. frondosa (mean pairwise divergence: 19%). Some studies suggest that C. xamachana and C. andromeda might be conspecific given their morphological and genetic similarities (Arai et al., 2017; Gamero-Mora, 2020; Holland et al., 2004; Jarms et al., 2019), but Gamero-Mora et al. (2022) treated these as separate species because of significant divergence and reciprocal monophyly on the basis of 28S ribosomal sequences; this was supported by Muffett & Miglietta (2023). Like C. xamachana, C. andromeda has subsequently been reported from the tropical and subtropical western Atlantic, Mediterranean Sea, the Indo-Pacific (including Australia, Gershwin, Zeidler & Davie, 2010; this study) and recently, the eastern Atlantic (Gueroun et al., 2024). For these two species, geography is unhelpful in species identification given that both can be sympatric in parts of their range as a result of both natural and artificial processes, and the additional possibility that other species may be present (Fitt et al., 2021; Muffett & Miglietta, 2023).

Specimens of a second species occurring in eastern Australia analysed here from Lake Illawarra, New South Wales, and Coombabah Creek, Queensland, are found to be closely related to specimens from Japan (Abboud, Daglio & Dawson, 2018), Papua New Guinea and Hawaii (Holland et al., 2004), reported as Cassiopea sp.3. While there is some genetic variation between Australian populations, including Lake Alexander, NT, and those from overseas (mean pairwise divergence: 6.7%), they are here considered a single species due to the pairwise divergence being lower than the minimum between closely related species recognised by Gamero-Mora et al. (2022) (i.e., 7.7% between C. xamachana and C. andromeda). Additionally, while Keable & Ahyong (2016) identified the population in Lake Illawarra as C. ndrosia, based on morphology and previous records using this name, the morphological features of these populations require comparison with those from the type locality of C. ndrosia, Suva Harbour, Fiji. In addition to more detailed morphological analysis, it is also recommended that future studies consider use of a wider suite of markers, especially those from the nuclear genome to allow for a more comprehensive and robust understanding of species distinctions and their relationships.

Morphometric analysis detected significant morphological differences between the two species occurring on the east coast of Australia and also Cassiopea sp.2 from Papua New Guinea. These consistent differences corroborate their previous treatment as separate species (Chiaverano, Bayha & Graham, 2016; Holland et al., 2004; Lessios, 2008), which is also indicated by the genetic analysis in this study. Three main features are significant in the morphometric separation of C. xamachana and Cassiopea sp.3 are: (1) the number of large appendages on the oral disc, which is much higher in Cassiopea sp.3 (at least 1 but up to 14 in medusae >10 cm), with C. xamachana having a maximum of two large appendages; (2) the oral arm branching pattern, which is usually alternating for C. xamachana, but a combination of alternating, bifurcating and pinnate for Cassiopea sp.3; (3) the length of the large appendage on the oral arm (average of 2.6 cm, which is 0.2 bell diameter, compared to 1.4 cm for Cassiopea sp.3, which is 0.12 bell diameter). There was some overlap on the nMDS plot, which represents interspecific variation.

Characters of the large appendages in Cassiopea are shown here to be useful and reliable in separating some species. They are believed to have two main functions. First, they provide additional surface area for zooxanthellae, and second, they contain nematocysts in clusters known as cassiosomes, which are used for feeding to capture prey or released in defence (Ames et al., 2020; Larson, 1997; Stampar et al., 2020). Historically, clear and consistent information about large appendages has not been included in descriptions of Cassiopea, and as a result there is no standardised terminology, leading to taxonomic confusion (Gamero-Mora, 2020). Whereas, the present study recorded variations in size, colour, and shape of large appendages within species, their distribution and the location of the longest large appendage proved particularly useful for comparing Australian populations of C. xamachana and Cassiopea sp.3.

The characters that we identify as consistent across C. xamachana populations along the east coast of Australia are also consistent with the original description of specimens from Jamaica, Caribbean Sea, by Bigelow (1892) and the revision of this species by Gamero-Mora (2020), respectively. These features include generally five lappets per paramere; ribbon shaped large appendages located along the oral arms, and one on the bifurcation of each branch. However, Bigelow (1892) and Gamero-Mora (2020) record 5–13 large appendages on the oral disc, which is far more than we found in Australian specimens. Conversely, Gamero-Mora (2020) also record variations between populations of C. xamachana, noting that the Atlantic Mexican population only has one appendage located on the oral disc if present at all, which is consistent with the Australian material observed here. Additionally, characters highlighted in the original description of C. maremetens by Gershwin, Zeidler & Davie (2010) (see Taxonomy section) and comparison with other species by Gamero-Mora (2020) are consistent with the distinguishing features identified in this study, including five lappets per paramere, up to two large appendages on the oral disc, and a leaf shaped large appendage on the bifurcation of the oral arm. While Gershwin, Zeidler & Davie (2010) also recorded the absence of large appendages on some of the paratypes of C. maremetens, those specimens are not from the type locality, but from localities within Moreton Bay and without molecular data, and so were not examined in this study—these might represent other species. As a result, molecular comparisons of specimens from these localities need to be made with known C. xamachana and Cassiopea sp.3 populations to corroborate the identification of these paratypes.

Keable & Ahyong (2016) identified the species occurring in Lake Illawarra as C. ndrosia; in comparison with the taxonomic revision by Gamero-Mora (2020) and the specimens in this study assigned to Cassiopea sp.3, some of the features are consistent with this identification as C. ndrosia, including the rhopalia (14–23) counts and the presence of large appendages on the oral disc, which are conversely absent from the oral arms. Despite morphological similarities between Cassiopea sp.3 and the description of C. ndrosia by Gamero-Mora (2020), a detailed morphological data and molecular data are not yet available for specimens definitively identified as C. ndrosia (type locality: Suva Harbour, Fiji). Therefore, both genetic and morphological comparisons need to be made between Cassiopea sp.3 from Australia, Japan, Hawaii, Papua New Guinea, and specimens from the type locality of C. ndrosia, Suva Harbour, Fiji, before a positive match for Cassiopea sp.3 can be made with an available taxonomic name.

Although further revisionary study of Cassiopea occurring in Australia is needed, present confirmation of the presence of C. xamachana in Australia is significant given the attention this species has received at other localities as an invasive species. The native range of C. xamachana (whether the tropical western Atlantic or the central western Pacific) warrants further study, but the minimal genetic diversity in eastern Australia (a single haplotype; Fig. 3) in addition to its apparently recent appearance there suggests that is a recent arrival, more so because Cassiopea are conspicuous wherever they occur. Certainly, the southward expansion by C. xamachana and its localised blooms indicate that it is currently invasive in New South Wales coastal lakes (Gershwin, Zeidler & Davie, 2010; Keable & Ahyong, 2016; Rowe et al., 2022b). As a probably new arrival, rather than a native species increasing its range, the expanding distribution records within Australia are of greater importance than previously recognised. Additionally, further study to determine vectors by which this species may be being translocated, including whether Cassiopea is naturally expanding its range, or whether it is human-mediated introductions.

Taxonomy

Order RHIZOSTOMEAE Cuvier, 1800

CASSIOPEIDAE Tilesius, 1831

Cassiopea Peron & Lesueur, 1810

Cassiopea xamachana Bigelow, 1892

Figures 8–10.

Figure 10 Live specimen of Cassiopea xamachana collected from Lake Macquarie, 6 May 2019. Specimen is 17.4 cm in diameter.

(A) Oral view. (B) Aboral view.

Cassiopea xamachana Bigelow, 1892: 212–221.—Bigelow, 1893: 301.—Bigelow, 1900: 191, figs A–L, pl. 31–38.—Mayer, 1910: 499–735, pl. 56–76.—Kramp, 1961.—Hummelinck, 1968: 1–57.—Holland et al., 2004: 1119.—Morandini et al., 2017: 321.—Ohdera et al., 2018: 1.—Gamero-Mora et al., 2019: 1.—Jarms et al., 2019: 504.

Cassiopea maremetens Gershwin, Zeidler & Davie, 2010: 91, fig. 6C–F.—Templeman & Kingsford, 2015: 1–8.—Epstein, Templeman & Kingsford, 2016: 340–346.—Jarms et al., 2019: 492.—Gamero-Mora, 2020: 113, fig 43.—Rowen, Templeman & Kingsford, 2017: 143–148.—McKenzie, Templeman & Kingsford, 2020:1–7.—Templeman, McKenzie & Kingsford, 2021: 1. [New synonymy]

Cassiopea cf. maremetens.—Keable & Ahyong, 2016: 26, figs 2E–H, 3C–D, 4D–F.

Cassiopea sp.—Rowe et al., 2022a: 2.—Rowe et al., 2022b: 1.

Cassiopea medusa.—Durieux et al., 2023: 9 [nomen nudum; not C. medusa Light, 1914, erroneous citation of Rowe et al. (2022a: 2) who indicate “Cassiopea medusae”, i.e., medusae of Cassiopea sp., subsequently identified here as C. xamachana].

?Cassiopea andromeda.—Stephenson, 1962: 94 [doubtfully C. andromeda Forskål, 1775].

Type material of C. maremetens examined. Holotype: QM G.326486, female (17.5 cm diameter), off Lamerough Canal, Lake Magellan, Pelican Waters, Queensland, Australia, 26°49′47″S, 153°6′36″E, D. Potter and G. Cranitch, 24 May 2007. Paratypes: QM G.6645, eight specimens (9.8, 10, 10.9, 10.3, 11, 11.5, 11.5, 11.5 cm), Mud Island, Moreton Bay, Queensland, Australia, 27°19′47″S, 153°15′0″E, C. Wallace, 4 August 1972. QM G.327932, two females (8.8, 17.2 cm), off Lamerough Canal, Lake Magellan, Pelican Waters, Queensland, Australia, 26°49′47″S, 153°6′36″E, D. Potter and G. Cranitch, 24 May 2007. QM G.327969, two females (11.9, 14.5 cm), estuary on NW side of Bentick Island, Queensland, Australia, 17°3′35″S, 139°29′24″E, P. Davie, 20 November 2002.

Additional material examined (all from Australia). Pelican Waters, Queensland: AM G.20068-20069, two females (15.5, 16.1 cm), 26°50′01″S, 153°06′44″E, C.E. Rowe, 10 March 2021. Gold Coast, Queensland: QM G.339123–339125, three specimens (5.9, 7.7, 12.1 cm), 28°3′S, 153°24′26″E, M. Ekins and I. Jamieson, 8 August 2019. QM G.339126–339130, five females (6.7, 7.2, 8, 10.3, 17.3 cm), 28°10′S, 153°24′37″E, M. Ekins and I. Jamieson, 8 August 2019. SeaWorld Culture, Queensland: AM G.18699–18701, three specimens (6.1, 6.5, 10.4 cm), most probably originating from a small inlet on southern end of South Stradbroke Island, Gold Coast Council Region, 25°55′3″S, 153°25′15″E, 22 January 2019. Wallis Lake, New South Wales: AM G.18137–18139, 13 specimens (6.4, 7.7, 7.7, 7.7, 9.2, 10.4, 10.9, 11.2, 11.3, 12.1, 12.7 cm), in channel splitting Godwin Island approximately one-third distance from southern shore, 32°11′45″S, 152°29′55″E, R. Pearce, 15 August 2014. AM G.18143–18156, 12 specimens (2.9, 3.6, 4.4, 5.2, 5.6, 7.2, 8.4, 8.8, 8.8, 9.4, 9.6, 9.7, 10.7 cm), behind Smugglers Cove Caravan Park, Pipers Creek, “The Keys”, 32°12′0″S, 152°30′39″E, R. Pearce, 18 September 2014. AM G.18181–18183, four specimens, inlet on south-east side of Mather Island, 32°11′26″S, 152°29′36″E, S.J. Keable and A.D. Hegedus, 28 April 2015. AM G.18184, one specimen, Pipers Creek within Smugglers Cove Caravan Park, 32°11′58″S, 152°30′39″E, S.J. Keable and A. Murray, 30 April 2015. AM G.18736–18755, 20 females (8.9, 11, 12, 12.5, 12.8, 13.1, 13.3, 15, 15.1, 15.1, 16.7, 17.3, 18.7, 19.1, 19.4, 19.4, 20.8, 20.9, 22.3, 24 cm), Pipers Creek within Smugglers Cove Caravan Park, 32°11′58″S, 152°30′39″E, S.J. Keable and C.E. Rowe, 10 May 2019. Lake Macquarie, New South Wales: AM G.18362–18365, 12 specimens (7.1, 10, 12.6, 13.1, 13.3, 13.4, 16.7, 17, 17.4, 17.5, 19.6, 19.7 cm), Lake Petite, 33°06′59″S, 151°32′04″E, S.J. Keable, D.J. Keable, S. Jones, D. Jones, and E.M. Keable, 4 June 2017. AM G.18428, one specimen (13.4 cm), Lake Petite, 33°07′00″S, 151°31′58″E, S.J. Keable and A. Hay, 9 February 2018. AM G.18528, 1 specimen (6.1 cm), Karignan Creek, 33°10′36″S, 151°34′03″E, C.E. Rowe and S.J. Keable, 16 May 2018. AM G.18711–18712, two specimens (11.9, 12.4 cm), Mannering Bay, 33°09′28″S, 151°31′41″E, C.E. Rowe and S.J. Keable, 7 March 2019. AM G.18716–18723, eight females (10.3, 14.3, 12, 15.5, 16.4, 17, 19.6, 20 cm), Karignan Creek, 33°10′32″S, 151°34′00″E, S.J. Keable and C.E. Rowe, 6 May 2019. AM G.18724–18728, five females (11.3, 12, 12.2, 12.9, 14.4 cm), creek north of Kilaben Creek, 33°01′43″S, 151°35′01″E, S.J. Keable and C.E. Rowe, 9 May 2019. AM G.18729–18735, seven females (14.2, 16.1, 17.2, 19.2, 19.7, 19.8, 20.9 cm), Lake Petite, 33°07′03″S, 151°32′06″E, S.J. Keable and C.E. Rowe, 9 May 2019.

Comparative material examined. Cassiopea sp.3: Northern Territory, Australia: AM G.17363 and G.17374, three specimens (6.3, 6.8, 6.8 cm), Lake Alexander, Darwin, 12°24′S, 130°49′E, M. Dawson, 15 November 2003. Queensland, Australia: AM G.20057–20061, five females (14.9, 15.4, 16.2, 16.3, 18.2 cm), Kangaroo Avenue, Coombabah Creek, Gold Coast, 27°05′26″S, 153°22′57″E, C.E. Rowe and J. Lawley, 8 March 2021. AM G.20064–20067, four specimens (2.3, 5.1, 9.9, 10.7 cm), SeaWorld culture originating from Coombabah Creek and South Stradbroke Island, Gold Coast Region, 25°55′3″S, 153°25′15″E, 9 March 2021. AM G.13568, three specimens (10.1, 12.2, 13.3 cm), off Hayman Island, Whitsunday Passage, 20°03′S, 148°53′E, JA. McNeill, 17 April 1934 (material referred to by Stiasny (1931) and Keable & Ahyong (2016) as C. ndrosia). New South Wales, Australia: AM G.18075, one specimen (9.2 cm), canal through jetties by the lake at Windang Road, Lake Illawarra, 34°31′36″S, 150°51′52″E, M. Cameron, 8 May 2013 (material referred to by Keable & Ahyong (2016) as C. ndrosia). AM G.20077, 11 specimens (3, 4.2, 5.4, 6.9, 7.2, 7.3, 8.2, 9.8, 10.5, 10.8, 15.1 cm), canal through jetties by the lake at Windang Road, Lake Illawarra 34°31′36″S, 150°51′52″E, S.J. Keable and C.E. Rowe, 1 June 2021. AM G.20076, two specimens (12.7, 12.8 cm), channel between rocky shore and sand spit, north-east of entrance to Cabbage Tree Basin, Port Hacking 34°04′44″S, 151°07′58–E, S.J. Keable and C.E. Rowe, 31 May 2021.

Cassiopea sp.2: Papua New Guinea: AM G.17370 and G.17385, six specimens (4.6, 7.3, 7.7 8.9, 9,10.5 cm), outer slope, west side of Mascot Channel mouth, 2°40′04″S, 150°25′58″E, P.L. Colin, 3 July 2003. AM G.17387, one specimen (5.2 cm), atoll −70 nautical miles south-south-east of Manus Island, Sherburne Reef, 3°19′59″S, 148°01′03"E, D. de Mara, 20 June 2002. Queensland, Australia: AM G.18344, one specimen (6.3 cm), north-east coast, Lizard Island, 14°40′01″S, 145°27′37″E, A. Hoggett, 21 February 2016.

Cassiopea andromeda: South Australia: SAM H3568, H3577–78 and H3581, four specimens (6.2, 6.2, 7.9, 9.7 cm), boat ramp, Garden Island Yacht Club, Garden Island, Angas Inlet, 34°48′11″S, 138°31′55″E, M. Bossley and A. Crowther, 4 May 2022.

Description. (Where variation present, value for holotype of C. maremetans given in brackets). Exumbrella: marginal outline circular; aboral surface mostly flat with small concavity; maximum diameter 24 cm, mean 11.5 cm (holotype: 17 cm); height 0.2–2.7 cm (0.4 cm), mean 0.9 cm. Colour pattern brown, yellow, green (greenish yellow). Markings include small white spots around bell margin, triangle facing towards each ocellus on live specimens (markings lost in preservation). Rhopalia 13–23 (19), mean 15, located in incised notches; ocelli present. Marginal lappets 32–120 (104), mean 83 depending on the condition of the specimen, with 3–7 per paramere (4–6), not always distinct, distal edge with rounded low lobes; 1–5 (3) velar lappets located between two ocular lappets. Ring canals 1 in total, raised.

Oral disc: 0.2–2 cm height, with a mean of 0.94 cm, 0.5–8.2 cm width, and with a mean of 3.88 cm, typically 1.6–5.4 × the bell diameter, with a mean of 0.4 × bell diameter. Subgenital pit circular or rhomboid, 0.1–0.9 cm width, with 0.65 cm mean, 0.09–0.58 × oral disc height, with 0.3 mean. Large appendages present on specimens >8 cm; 1 or 2 in centre of oral disc (1), fusiform, 0.7–3.4 cm in length (2.5 cm), averaging 1.3 cm, 0.17–0.67 × oral disc width, but typically 0.25 × oral disc width, width of large appendage 0.06–0.4 × length of large appendage (0.4), typically 0.27 length of large appendage.

Oral arms: 7–10 (8), usually 8 (except AM G.18733, G.18728, QM G.327932, G.6645, G.32769). 0.9–14.5 cm in length (12.3) with mean of 6.38 cm, 0.75 to 1.54 × bell radius (1.5 × bell radius) with mean 1.2 × bell radius, extending radially beyond bell margin, cylindrical in shape. Lateral branches 3–7 per oral arm (4 or 5), cylindrical; alternating branching pattern; secondary lateral branches with alternating branching pattern of equal strength. Trunk of oral arm ending in unequal bifurcation. Small appendages numerous, flat, fusiform, transparent, distributed evenly along oral surface of oral arm and lateral branches. Large appendages present; largest located at bifurcation of oral arm, 0.25.0 cm in length (1.7 cm) with mean of 2.50 cm, 0.07–0.63 × oral arm length with mean 0.3 × oral arm length, and width 0.05–0.36 × length, but typically 0.22 length. Number of large appendages on oral arm varying with size, with two large appendages near distal bifurcation in smaller specimens (e.g., AM G.18754 (11 cm), G.18744 (12.8 cm), G.18723 (12 cm)) and up to 17 evenly distributed along the oral arm in larger specimens (e.g., AM G.18365 (17 cm) and G.18362 (19.7 cm), holotype QM G.326486 with 2, 1 located at distal bifurcation and other half-way along oral arm, but this may be an artefact of damage, see Remarks), always present at distal bifurcation, fusiform.

Gonads: Always visible, cruciform, as mature gonads indicating sex, but definition varying with size and condition of preserved specimen; developed in specimens >10 cm in diameter.

Remarks

The diagnostic characters of Australian specimens, including the type material of C. maremetens and additional material from the type locality, are indistinguishable from those of C. xamachana from the western Atlantic (Bigelow, 1892). These include the number of lappets per paramere (4–6) and their indistinct, rounded shape; the alternating lateral branches along the oral arm, which end with an unequal bifurcation; and the longest large appendage located at the bifurcation in the oral arm, with the number of large appendages on the oral arm increasing with size (1–17), with maximum two located at the centre of the oral disc. These features are consistent across all specimens examined from Wallis Lake, Lake Macquarie and Pelican Waters, which are also genetically invariant in COI. Additionally, the morphometric analysis supports these as consistent features distinguishing C. xamachana from Cassiopea sp.3. As such, C. maremetens and C. xamachana are indistinguishable and we recognise the former as a junior synonym of the latter.

Variation in Australian C. xamachana

No significant morphological differences were detected between C. xamachana occurring along the east coast of Australia. However, small variations were evident, such as the number of lappets per paramere (within and between specimens), the number of oral arms and lateral branches, and, in the larger specimens, whether one or two large appendages are present in the centre of the oral disc.

Our examination found some morphological differences between specimens studied and the original description of C. maremetens by Gershwin, Zeidler & Davie (2010). For example, the original diagnosis and the holotype description indicate absence of ocelli, but these were found to be present in the holotype, additional material examined in this study and specimens examined by Keable & Ahyong (2016) from Wallis Lake. Additionally, the original diagnosis recorded one large appendage at the base of the oral arm pairs, and one at the distal bifurcation of each arm. Although the location of the large appendage in the holotype is consistent with our observations, other specimens we examined had a much higher number of large appendages on the oral arms, with up to 17 recorded on larger specimens. However, we note that the holotype is in relatively fragile condition, so some large appendages may have been damaged or lost. Additionally, the original description is based on specimens from multiple locations in the Moreton Bay region, including the paratypes QM G.327969 from Bentick Island, and QM G.6445 from Mud Island. These specimens are also in poor condition so many characters cannot be accurately assessed, but we were still able to observe numerous large appendages around the oral disc. This is a feature of Cassiopea sp.3, which also occurs in the Moreton Bay region. Therefore, the paratype series of C. maremetens may also include misidentified specimens of Cassiopea sp.3. As a result, these localities should be resampled so morphology and genetic information can be used to confirm the identification of these specimens.

Comparison of Australian and overseas C. xamachana

Cassiopea xamachana was first described from Kingston Harbour, Jamaica (Bigelow, 1892), and has since been reported widely around the world, most commonly in the Western Atlantic from Florida (Verde & McCloskey, 1998) south to Brazil (Gamero-Mora et al., 2019), as well as several Indo-West Pacific localities including Hawaii (Holland et al., 2004) and Palau (Arai et al., 2017; Jarms et al., 2019). The species was re-described by Gamero-Mora (2020), who set forth plans to designate a neotype from Fort Jefferson, Dry Tortugas, Florida Keys, Florida, USA, as the original type material is lost. Both the original description and redescription note that the characteristic features of this species include: five lappets per paramere, which are obtuse and separated by grooves in the bell margin; there are usually 16 rhopalia; four pairs of oral arms that contain 10–15 lateral branches that are alternating and contain additional secondary branches; and large appendages located in the axil of each oral arm, whose length can be up to one-fourth the bell diameter. Additionally, they record that the centre of the oral disc contains 5–13 of the large appendages, but in the prospective neotype, they are also scattered along the oral arms. The number and location of the large appendages on the oral arms of the prospective neotype, and the number of lappets and rhopalia are consistent with distinguishing features identified in C. maremetens. These similarities, combined with our wider genetic and morphometric analysis of Australian material corroborate the conclusion that C. maremetens is conspecific with C. xamachana.

It has been reported that there are morphological variations, even in clonal populations of Cassiopea (Hummelinck, 1968; Mayer, 1910; Morandini et al., 2017; Stampar et al., 2020). Therefore, some morphological variation among other populations of C. xamachana can be expected, especially given the wide distribution of the species. This includes the absence of grooves occurring between the lappets, which in the Australian specimens are rounded and connected. Additionally, the original description of C. xamachana indicates that the specimens have 5–13 large appendages on the oral disc, but the Australian specimens have a maximum of two. However, Gamero-Mora (2020) concluded that there are morphological differences in the Mexican population of C. xamachana, noting specimens had 6–10 oral arms, that were up to 1.6 × the bell radius, and the number of large appendages range from zero to 25 and are either on the distal part or scattered over the oral arms, with at least one located on the oral disc, which resembles the condition of specimens from Wallis Lake, Lake Macquarie and Pelican Waters, Australia.

Comparison of C. xamachana to other species of Cassiopea in Australia

The identification of C. ndrosia reported by Stiasny (1931) and Keable & Ahyong (2016) needs to be examined more closely to confirm their identification with sequences matched to morphology (see Discussion). Therefore, in this study, specimens have been designated Cassiopea sp.3 according to previous placement of molecular sequences (Holland et al., 2004). Key distinguishing features that separate C. xamachana from Cassiopea sp.3 include fewer large appendages on the oral disc (1 or 2 compared to 6), the location of the large appendages (oral arms compared to oral disc), length of the large appendage on the oral arm (usually 0.2 the bell diameter compared to 1.2), oral arm length (usually 0.57 the bell diameter compared to 0.67), lappets per paramere (usually 5 compared to 4), and the location of the large appendages on the oral arm (evenly distributed and always at the bifurcation, compared to sometimes at the bifurcation, but often absent), and the oral arm branching pattern, which is usually alternating for C. xamachana, but a combination of alternating, bifurcating and pinnate for Cassiopea sp.3.

Further comparisons need to be made with the South Australian population in the vicinity of Angas Inlet. Although Southcott (1982) reported Angas Inlet specimens as C. ndrosia, our results identified C. andromeda, which is closely related to C. xamachana and morphologically similar, including sharing these features: a large appendage on the fork of the oral arms; 1–2 central large appendages on the oral disc; and an alternating oral arm branching pattern. However, the specimens are juvenile (<10 cm), so many of the other key characters useful to distinguish species of Cassiopea were not fully developed. Given Southcott’s (1982) identification of material from Angas Inlet as C. ndrosia, the potential presence of another species, formerly or currently in the area, needs to be considered.

Conclusion

The species of Cassiopea occurring in Lake Macquarie, Wallis Lake and Pelican Waters (type locality of C. maremetens) is genetically and morphologically referable to C. xamachana. We provide a detailed description of Australian C. xamachana, including a comparison with the type and topotypic material of C. maremetens, synonymised herein. Additionally, we provide evidence that a second species, C. sp.3 (previously identified as C. ndrosia), is expanding its range south along the east coast of Australia and identify the morphological features that distinguish it from C. xamachana. As the two species continue to expand their range southwards in eastern Australia, which is expected under climate change scenarios (Rowe et al., 2022a), these diagnostic features will facilitate their identification, especially in monitoring and management of their invasion front.

Supplemental Information

Supplemental Information 1 COI sequences used in this study for analysis of Australian Cassiopea populations..

Extracted from material made available by SAM, AM, ReefHQ or QM, or obtained from GenBank.

Supplemental Information 2 Global COI sequences used in this study.

Sequences sourced both from this study and GenBank.

Supplemental Information 3 Specimens examined for morphometric comparison.

Supplemental Information 4 Pairwise distances from the Australian COI alignments.

Supplemental Information 5 Pairwise distances from the Global COI alignments.

Supplemental Information 6 Cluster dendogram of all specimens of Cassiopea examined.

Red lines indicate morphologically homogeneous clusters detected by Ward Hierarchical Clustering and the k-means of the specimens. Labels indicate species and their registration number. All specimens from the AM, except for the Holotype (QM G.326486).

Supplemental Information 7 Cluster dendograms of the putative populations of Cassiopea.

A) Cassiopea xamachana Bigelow, 1892. B) Cassiopea sp.3. Red lines indicate morphologically homogeneous clusters detected by Ward Hierarchical Clustering and the k- means of the specimens. Labels indicate the population and their AM registration number.

Supplemental Information 8 Morphological Measurements.

Data used for the morphometric analysis.

Supplemental Information 9 Data used in R for the morphometric analysis.

Rhopalia, bell height, total height, oral disc width, oral disc height, arm vesicles, arm vesicle width and central vesicle length are divided by bell diameter. Z-score calculated for lappets, subgenital pit, lateral branches, and central vesicle width. For distinct lappets, 1 represents no, 2-yes. Lappet shape, 1-rectangular and blunt, 2- rectangular and round, 3- rounded, 4- square, 5- square and blunt, 6- square and round. Arm longer than bell 1- no, 2- same, 3- sometimes, 4-yes. Arm Shape 1-flat, 2-cylinder, 3-traingle. Branching pattern 1- alternating, 2- alternating and bifurcating, 3- bifurcating, 4- pinnate. Fork strength 1- equal, 2- one side stronger. Arm vesicle distribution 1- absent, 2- base, 3- base and fork, 4- evenly distributed, 5- fork, 6- tip of arm. Large vesicle location 1-central, 2-equal, 3-fork. Vesical shape 1-leaf, 2- test tube and leaf, 3- test tube.

Supplemental Information 10 R Code for Morphometric Analysis.

The authors would like to thank Dr. Merrick Ekins, and Dr. Andrea Crowther, the Collection Managers at Queensland Museum and South Australian Museum respectively, for the loan of specimens. Additionally, we would like to thank Alex Hegedus, Professor Kylie Pitt, Dr. Jonathan Lawley, and Dr. Mike Bossley for their assistance in collecting specimens for this study. We would also like to thank Dr. Edgar Gamero-Mora for guidance and discussions about upside-down jellyfish genetics and morphology, and thank you to the editor and reviewers for their insightful comments.

Additional Information and Declarations

Competing Interests

The authors declare that they have no competing interests.

Author Contributions

Claire E. Rowe conceived and designed the experiments, performed the experiments, analyzed the data, prepared figures and/or tables, authored or reviewed drafts of the article, and approved the final draft.

Shane T. Ahyong conceived and designed the experiments, performed the experiments, analyzed the data, authored or reviewed drafts of the article, and approved the final draft.

Will F. Figueira conceived and designed the experiments, performed the experiments, analyzed the data, prepared figures and/or tables, authored or reviewed drafts of the article, and approved the final draft.

Ingo Burghardt conceived and designed the experiments, performed the experiments, analyzed the data, authored or reviewed drafts of the article, and approved the final draft.

Stephen J. Keable conceived and designed the experiments, performed the experiments, analyzed the data, authored or reviewed drafts of the article, and approved the final draft.

Field Study Permissions

The following information was supplied relating to field study approvals (i.e., approving body and any reference numbers):

Permit received from NSW Department of Primary Industries.

DNA Deposition

The following information was supplied regarding the deposition of DNA sequences:

The DNA sequences are available at GenBank: PV365289 to PV365299, PV365302, PV365304 to PV365313, PV365315 to PV365323.

Data Availability

The following information was supplied regarding data availability:

The morphometric data and R code are available in the Supplemental Files.

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
