# Peer review of "Identification of Cassiopea sp. in Lake Macquarie, Australia and revision of the taxonomic status of Cassiopea maremetens Gershwin, Zeidler & Davie, 2010 (Cnidaria: Scyphozoa: Cassiopeidae)"

_PeerJ, doi:10.7717/peerj.19669_

## Round 0.1 · original submission · Minor Revisions

Dear author,

Please read the comments from the four referees and make the changes in the MS and give reasons if you disagree with them for not making changes.

kind regards

·

Basic reporting

The research paper by Rowe and co-author is well written and has a good flow. I found a couple of spelling mistakes, check my comments in the annotated PDF.
The introduction is well written and show context. However, I believe some concepts should be better explained and some examples given. For instance, the concept of alien species should be reported. Cassiopea andromeda for example is an alien species in the mediterranean sea and in Brazil. There are 2 major concepts here: invasiveness but also alien species. Some species can be indigenous and invasive, some only indigenous, some just alien, some alien and invasive. Alien species are one of the major threat for biodiversity and if alien species are invasive, impacts on ecosystems could be even bigger. Cassiopea is sometimes just an indigenous invader but sometimes is an alien invaders (e.g., in the Mediterranean Sea). So I strongly believe that the authors should better explain this concepts in the introduction.
Additionally, I encourage the authors to avoid citing too much reviews but instead cite the original papers that conducted the work.
I like the figures but Figure 1 and 2 look a bit pixelated. I also believe that figure 2 could be more explicatory (see comment in PDF).
The R codes and raw data are presented as supplementary. The link for deposited sequences is missing but I saw that they are still pending so all the raw data were supplied.

Experimental design

The Research presented in the manuscript by Rowe et al. falls within the scope of peer J and the specimens’ analysis was carried out with high rigor and details.
The research conducted by Rowe et al. is relevant and meaningful and helps clarifying the occur problem of taxonomic ambiguity.
The methodology is well described and replicable.

Validity of the findings

Cassiopea species are very difficult to distinguish solely based on morphology and this paper carries out a very in-depth analysis of morphology integrated with molecular data of Cassiopea in Australia.
The data provided in the paper by Rowe et al. are robust, well presented/illustrated, analyzed, and the results support the research question and the aim of clarifying Cassiopea taxonomy

Reviewer 2 ·

Basic reporting

Line 50: Please clarify the distinction between “sedentary” and “semi-sessile” in the context of Cassiopea. While they are typically stationary, their ability to produce cassiosomes, continuously pulsate their bells, and actively reproduce suggests a level of activity that may not fit the traditional definition of sedentary. Providing a clear justification for your chosen terminology would strengthen this section.
Line 51: The term “extended above” may be ambiguous. Consider specifying that the oral arms are positioned upward, facing the water column, to provide clearer anatomical context.
Line 53: Please cite at least two more references for this sentence.
Line 58: Add Morandini et al., 2017 to the citations
Line 63: Can you provide more specificity regarding the factors influencing Cassiopea blooms? While some studies suggest ocean acidification and increasing temperatures may not favor their proliferation, human impacts such as coastal development have been shown to contribute to their blooms (see Stoner et al. 2014 and López-Figueroa, 2023). Incorporating this perspective could provide a more balanced discussion.
Line 74: You mention that the species exhibits great phenotypic plasticity, yet it maintains a relatively conservative morphology. Could you clarify what specific factors contribute to this morphological stability despite its plasticity? For example, are there genetic, ecological, or functional constraints that limit morphological variation?
Line 131: Consider providing a table that includes key details such as location, coordinates, habitat description, and sampling date. Additionally, assigning reference letters or numbers to each site could help readers easily cross-reference these locations with the map, improving clarity and accessibility of the data.
Lines 210-212: These sentences could be condensed into one, seems redundant as is.
Line 243: what categorical variables? Suggestion “…categorical variables (i.e., X, Y Z).
Line 244: please verify version number of R
Line 252: add comma between species and cluster
Line 257: Remove last sentence “All statistical analyses were completed in R (v2021.09.01).” It was already mentioned about analyses were performed using R
Line 261: reconsider a more specific subsection. It was previously established that the location of collection was Australia, maybe change to what the paragraph is informing.
Line 276: reconsider subsection title. Comparison results or something along the lines of what it is described in this section to match the headings/order of results.
Line 325: Consider López-Figueroa 2024, their literature review data to support this statement about the need for molecular analysis.
Line 358: Gamero-Mora missing the hyphen
Line 375: remove Holland et al. 2004 as it has been mentioned right before.

Experimental design

This study represents a significant advancement in Cassiopea research, providing a strong model for future studies on species identification and distribution dynamics. The integration of genetic barcoding and morphometric analyses strengthens the reliability of species delineation, offering clarity on long-standing taxonomic uncertainties. Notably, the reassessment of C. maremetens as a junior synonym of C. xamachana marks an important step in refining our understanding of Cassiopea biogeography. The methodological rigor and depth of discussion in this manuscript make it a vital contribution to the field, setting a strong foundation for future studies on Cassiopea ecology, distribution, and potential impacts in new environments.

Validity of the findings

While the study highlights the range expansion of C. xamachana in Australia, the concept of Cassiopea as an ‘invasive’ species warrants further discussion. Given that Cassiopea populations have been present in certain locations for extended periods, phytogeographic analyses would be valuable in differentiating natural range expansions from human-mediated introductions.

Additional comments

To improve the usability and reproducibility of the code, consider adding comments throughout to guide users through each step of the analysis. Some sections, such as data preparation, clustering, and visualization, could benefit from brief explanations of their purpose and expected outputs. For instance, standardize variable names, and clarify data inputs/outputs. Additionally, specifying the structure and format of the input data files (e.g., required columns, data types) would enhance clarity and ensure that others can easily apply the code to their own datasets. If feasible, making the script available on GitHub with a README file explaining the workflow and dependencies would further improve accessibility and long-term usability.

·

Basic reporting

Yes all good in this regard.

Experimental design

All good in terms of experimental design for this type of study however additional genetic data would have been preferred.

Validity of the findings

Some overinterpretation of the findings apparent - see additional comments below for more information.

Additional comments

Thank you for the opportunity to review this integrated study on an intriguing group of Scyphozoans, a subject long overdue for further taxonomic research. The study has a modest objective of determining the identification of one population of Cassiopea in Lake Macquarie. In doing so, it presents compelling evidence supporting the synonymization of a currently accepted species.

Overall, the paper is well written. However, I found the second sentence of the abstract slightly awkward and suggest rewording it for better clarity. Similarly, the beginning of the third sentence should be revised to avoid starting with "These."

The introduction is strong but slightly brief. When discussing invasive species (lines 53–55), it would be helpful to provide more detail on the number of species considered invasive, specify which ones, and describe the novel environments they are expanding into.

The word ‘via’ seems missing in line 56.

Could all info from lines 66-73 be presented in a table with distribution info or a map showing current understanding of species distributions?

Line 90 – was the Holland study on a single gene? And which one?

Could all info from lines 131-142 be presented in a table?

My main concern with this study is that the molecular systematic interpretation relies solely on a single, relatively short (500 bp) gene that is known to be conserved. The authors do not address the adequacy of this gene for taxonomic purposes, either in marine invertebrates generally or in Scyphozoans specifically. Providing this context is crucial, as it would help justify the molecular approach and clarify its limitations in resolving species boundaries. I strongly recommend discussing this aspect to strengthen the study's taxonomic conclusions.

It is excellent to see that the authors have identified at least three distinguishing morphological characters or character states. To enhance clarity and support their findings, it would be helpful to include additional diagrams or images in the supplementary material. This would better visualize the character states described on lines 210–230, particularly those found to be significantly different. Providing these visual references would strengthen the morphological analysis and improve the study’s accessibility to readers.

My other main concern is the lack of bootstrap support for key nodes in the Australian and global phylogenetic reconstructions. It states on line 265 that Lake Mac specimens etc are ‘distinct’ from Lake Illawarra etc… but there is no bootstrap support value listed. Is this because the support value fell below 70? Or is this some sort of annotation error? If it’s the former – that means there is no support for this node being split from the Lizard I sample or possibly others above it – hence the interpretation here needs to be reined in.

Similarly for the global tree, there is no support for numerous of the important nodes in relation to putative sp 2 and sp. 3 (lines 283-285). So, I advise the authors to be more direct in saying that there is no support and add to the figure captions that only support values above 70% are shown (if that is the case).

Also, wrt morphometric analysis and dMDS – line 292 states there is ‘clearly separated groups’ but that is not entirely the case based on a few individuals as discussed further down. So again, I would request the authors go through and tone down the interpretation in places and make it clear where robust interpretations can be made and where it cannot.

Figures discussed in lines 305-311 in relation to cluster analysis would be better placed in the supps.
I see in line 313 you use the word ‘putative’ and that is a good word to use for the case of sp. 2 and sp.3 so I would encourage further use of that.

In the discussion important info is introduced about a neotype of C. xamachana being included – is that Florida Keys sequence in the tree from this neotype? If so, please add mention of that to the figure key and annotate with * (or similar).

It seems relatively convincing that C. maretmetens is a js of C. xamachana assuming the sequence in the tree is of the neotype. If it is not, then it may be too soon to make that conclusion.

I agree with all the discussion that more work needs to be done to resolve sp 2 & 2 and C. ndrosia and other putative spp. But can you say something more about what other molecular approaches may be useful in this regard in the future?

Also I found the discussion of the potential ‘invasiveness’ of C. xamachana as a bit superficial. Why do you think this is a new arrival? Rather than a native species expanding its range if it naturally occurs in other parts of the Western Pacific (aka Palau) or are you suggesting it is invasive to the wider Western Pacific?? I understand perhaps it is not easy to answer this question, but how could that potentially be resolved in the future? Seems like if it is invasive, this is really important biosecurity issue and further discussion is warranted.

Lastly, the quality of phylogenetic tree figures is pretty poor – can these be improved for clarity including increasing font size for all text and better annotating with colours or other panels on the RHS to show species groupings or other important information that will help with the interpretation.

·

Basic reporting

The manuscript is an important contribution to the knowledge of jellyfish diversity. It present relevant data on the identification of a previously endemic species, which based on new results (detailed morphology and wide molecular analysis) proved to be a non-indigenous species originating from the Caribbean. The structure of the text is very good and the language is appropriate and clear. All the important literature is mentioned, both old and new ones.

Experimental design

There is no experimental design properly to be evaluated. The study can be considered a test of hypothesis in order to validate C. maremetens as a distinct species. The results presented state that it is synonymous to C. xamachana, first described for Jamaica. The work is original in testing and checking the validity of the species. The problem is well defined, and the objectives were clearly stated and achieved with the results. The methods employed are adequate and the results can be replicated.

Validity of the findings

The results can be considered new because they provide further details on a previously described species. The authors are successful in providing both morphological and molecular data that corroborate the distinction of Australian specimens of Cassiopea, but are careful regarding another species not widely sampled. In general the manuscript helps in advancing the knowledge of the genus Cassiopea, largely know as quite difficult to identify. It is important in the general sense because it can help to monitor the expanding range of the genus in Australia. All presented data are clear and support the discussion and conclusion.

Additional comments

The text is very good and the authors are greeted to help advancing the systematics of the genus Cassiopea. Small comments and suggestions performed directly on the PDF manuscript file.

---

## Round 0.2 · accepted · Accept

I have read the revised version of this paper and agree with the reviewers in that it is ready to be accepted for publication.

·

Basic reporting

the paper is well written and clear, and it is well organized. Raw data and R codes are shared and accessible.
the authors have included sufficient literature to give context and the paper reports interesting results

Experimental design

The primary research falls within the scope of PeerJ and the idea behind the paper is well presented. The authors did a really good and rigorous job in describing a species, bringing very useful new knowledge on the genus. methods are very well described and enough to be replicated

Validity of the findings

the paper will be important and useful for the Cassiopea community, for taxonomists and jellyfish experts.

it covers a topic that because of its ambiguity is very debated: Cassiopea taxonomy. the paper brings relevant and interesting new information on this topic.

Additional comments

I really enjoyed reading the manuscript and I believe it is now ready for publication.

·

Basic reporting

I am happy with the revised version

Experimental design

All good

Validity of the findings

Findings are substantiated based on the reviewer responses and changes.

Additional comments

no comment

·

Basic reporting

The authors followed the suggestions of the reviewers. No further comments.

Experimental design

The authors followed the suggestions of the reviewers. No further comments.

Validity of the findings

The authors followed the suggestions of the reviewers. No further comments.

Additional comments

The authors followed the suggestions of the reviewers. No further comments.